# Myalgic Encephalomyelitis/Chronic Fatigue Syndrome: A Neurological Entity?

**DOI:** 10.3390/medicina57101030

**Published:** 2021-09-27

**Authors:** Iñigo Murga Gandasegui, Larraitz Aranburu Laka, Pascual-Ángel Gargiulo, Juan-Carlos Gómez-Esteban, José-Vicente Lafuente Sánchez

**Affiliations:** 1LaNCE-Neuropharm Group, Neuroscience Department, University of the Basque Country (UPV-EHU), 48940 Leioa, Bizkaia, Spain; larraitz.aranburu@ehu.eus (L.A.L.); juancarlos.gomez@ehu.eus (J.-C.G.-E.); josevicente.lafuente@ehu.eus (J.-V.L.S.); 2Experimental Psychology Laboratory, CONICET, Department Pathology, Universidad Nacional de Cuyo, Mendoza 5500, Argentina; gargiulo@lab.cricyt.edu.ar; 3Neurodegenerative Disease Group, Biocruces Research Institute, 48903 Barakaldo, Bizkaia, Spain

**Keywords:** Myalgic Encephalomyelitis (ME), Chronic Fatigue Syndrome (CFS), neuroimaging, dysautonomia

## Abstract

Myalgic Encephalomyelitis/Chronic Fatigue Syndrome (ME/CFS) is a disorder of unknown physiopathology with multisystemic repercussions, framed in ICD-11 under the heading of neurology (8E49). There is no specific test to support its clinical diagnosis. Our objective is to review the evidence in neuroimaging and dysautonomia evaluation in order to support the neurological involvement and to find biomarkers serving to identify and/or monitor the pathology. The symptoms typically appear acutely, although they can develop progressively over years; an essential trait for diagnosis is “central” fatigue together with physical and/or mental exhaustion after a small effort. Neuroimaging reveals various morphological, connectivity, metabolic, and functional alterations of low specificity, which can serve to complement the neurological study of the patient. The COMPASS-31 questionnaire is a useful tool to triage patients under suspect of dysautonomia, at which point they may be redirected for deeper evaluation. Recently, alterations in heart rate variability, the Valsalva maneuver, and the tilt table test, together with the presence of serum autoantibodies against adrenergic, cholinergic, and serotonin receptors were shown in a subgroup of patients. This approach provides a way to identify patient phenotypes. Broader studies are needed to establish the level of sensitivity and specificity necessary for their validation. Neuroimaging contributes scarcely to the diagnosis, and this depends on the identification of specific changes. On the other hand, dysautonomia studies, carried out in specialized units, are highly promising in order to support the diagnosis and to identify potential biomarkers. ME/CFS orients towards a functional pathology that mainly involves the autonomic nervous system, although not exclusively.

## 1. Introduction

Myalgic Encephalomyelitis/Chronic Fatigue Syndrome (ME/CFS) refers to the evolution and presence of severe, debilitating, and idiopathic chronic fatigue for more than 6 months, and is associated with other minor criteria, such as sleep disturbances, cognitive disorders, post-exertional discomfort, or pain [1].

The prevalence of chronic fatigue in developed countries is estimated to be around 20% [2], and 33% in Japan [3], with ME/CFS assuming only a small part.

Since 2008, Spain has had a consensus document for this syndrome, sponsored by the Institute for Health Carlos III and the Spanish Society for Neurology (SEN). It refers to a minimum of 0.1% [4] of the population affected by the syndrome, although the ranges are highly dispersed (0.0052 to 6.40%) depending on authors and methodology [5].

This pathology appears to be framed among neurological diseases in the International Classification of Diseases (ICD-11: 8E49) [6], even though it has numerous multisystemic repercussions. There is a medical debate about the nature of this entity. A survey conducted by the Association of British Neurologists in the United Kingdom, the birthplace of Myalgic Encephalomyelitis (ME), reported that 84% of the 351 neurologists surveyed indicated the pathology cannot be considered in the usual “neurological” sense, despite knowing its neurological classification [7].

ME/CFS affects both sexes, but more frequently women (4:1 ratio, women/men) [8,9], at any age (between 11 and 69 years), and mostly those of Caucasian ethnicity [10,11]. Direct and indirect costs are extremely high; in the United States alone, the cost was estimated to be USD 18–24 billion [12].

It is a pathology for which an etiopathogenesis or pathophysiology that clarifies the underlying mechanisms has not yet been established, referring to it as a neuro-immune-endocrine dysfunction, with an exclusively clinical diagnosis [13].

The nomenclature, classification, and diagnostic criteria have undergone various changes [14,15]. The most recent change took place in 2015 when the US National Academy of Medicine proposed the name of Systemic Exertion Intolerance Disease [16], a hybrid between the criteria for CFS (detailed by Fukuda, 1994) and ME (detailed by Carruthers, 2011), which does not solve the lack of definition of the syndrome.

ME/CFS presents features in common with other pathologies that lead to “central” fatigue, exercise intolerance, cognitive alterations, and the need for prolonged rest. Infections, muscle weakness, and the presence of dysautonomia symptoms can occur in Multiple Sclerosis [17], Parkinson’s disease, and other neurodegenerative disorders, as well as in non-psychotic major depression, a condition with which it can coexist with psychosomatic disorders, which are not excluded by the Fukuda criterion, which is the most frequently used criterion. Sometimes it is difficult to differentiate this process when symptoms are combined. Likewise, it is crucial to find differential diagnoses with pathologies, such as the Postural Orthostatic Tachycardia Syndrome (POTS).

Our aim is to present an updated vision of the changes that can be observed in ME/CFS by neuroimaging and in the evaluation of dysautonomia in order to find indicators or biomarkers that could serve to establish a suspected diagnosis, a positive identification, and/or for the follow-up of its evolution.

## 2. Development

The symptoms typically appear acutely, although they can develop progressively over years. The “central” fatigue together with physical and/or mental exhaustion after a small effort, sometimes minimal, is an essential trait for diagnosis [18].

The mechanisms of “central” fatigue are still unknown, including the brain areas where such information is processed. The fatigue is defined by “A sustained feeling of tiredness, which is not directly related to physical activity, although that can worsen it disproportionately, and additionally, does not improve with rest. Patients feel tired in the morning and experience an inability to perform any activity, being responsible for physical and cognitive weakness and appearing as the integration of emotions, volition, cognition and motility” [19].

Therefore, “central” fatigue would be secondary to the interrelation of different external and internal stimuli, involving cognitive, emotional, motor, and sensory factors [20]. In the genesis of this fatigue, areas from the prefrontal cortex and basal nuclei have been involved, pointing to dopamine as a relevant neurotransmitter [21].

The existence of a “fatigue network” homologous to the “pain network” (neuromatrix) is yet to be determined, but the circuits for executive functions (planning, sequencing, anticipation, reasoning, flexibility, etc.) and cognitive control (attention, working memory, and inhibition) [22] are postulated as potential biomarkers. In this sense, the hypothalamus acquires a special relevance as a key organizer to understand the homeostatic energy balance in those neurological diseases with “central” fatigue [23].

Sleep patterns were altered, for instance, referring to a “non-restorative” sleep (patients wake up too exhausted to perform daily activities). The mechanisms of this alteration have not been clarified but point to disorders in chronobiology (sleep/wake rhythms), where the retino-hypothalamic bundle, suprachiasmatic nucleus (hypothalamus), pineal gland, brain stem, and their connections play an essential role in the adjustment of the biological clock [24,25,26] and in understanding this neuro-metabolic disorder. Frequently, patients refer to a state described as mental fog or “brain fog”, which is characterized by sluggish, fuzzy, “not sharp” thinking.

The diagnostic criteria mostly used in the different studies are those enunciated by Fukuda et al. [1]. These raise some definition problems. On the one hand, it leaves the door open to various psychiatric pathologies, such as personality disorders or psychosomatic ones. On the other hand, it excludes those patients who do not suffer from any type of pain. The Fukuda criteria were branded as “ambiguous” in 2003 by Reeves et al. [27]. In such a way, Carruthers et al. (2011) advised abandoning them and using their proposed criteria as a result of a broader “international consensus” [18].

Another frequent deficit of many publications is the absence of a list for the comorbidities that usually accompany these patients (musculoskeletal, endocrine, autoimmune, dysautonomia, etc.). It is not a minor point, rather it is of enormous relevance when it comes to enabling the establishment of clinical subgroups. The US National Academy of Medicine strongly recommends, in order to better address the syndrome, a characterization of clinical phenotypes (CFS + autoimmune disease, CFS + Fibromyalgia, CFS + POTS, CFS + anxiety) [16].

## 3. Neuroimaging

Neuroimaging techniques, mainly magnetic resonance imaging (MRI), allow for the study of the morphology, metabolism (by spectroscopy), and anatomical (by tractography) and functional connectivity (by BOLD signal) of the different brain areas. Advances in MRI have allowed the growth of knowledge and enhanced interest of researchers in this field. MRI is currently preponderant over other techniques. We must emphasize that we are facing a functional pathology without evidence of a morpho-molecular substrate.

Studies of the brain volume using VBM (Voxel-based morphometry) show some regional differences for both gray and white matter [28,29,30], but there are no significant differences in respect to controls.

Studies of structural connectivity (tractography) using DTI (Diffusion tensor imaging) reflect changes in the white matter, which may or may not be reversible, for example, in the internal capsule or in the prefrontal area to compensate the dysfunction of the ascending reticular pathways [31]. Using DTI, Zeineh et al. (2015) proposed the arcuate fascicle, commonly involved in language functions and word learning, as a potential biomarker for the diagnosis and monitoring of ME/CFS [32]. No other group has thus far reported another study that corroborates or refutes this proposal.

Functional connectivity allows the mapping of synchronous and asynchronous activation in different brain areas. For this issue, fluctuations of the BOLD (blood oxygen level-dependent) signal are studied. BOLD is the more common technique used to explore functional connectivity, although it can be also studied using the arterial spin labeling (ASL) technique. Several authors [33,34,35,36] document variations in neural activity at rest and after slight activity, either an increase or decrease, in brain areas involved in executive and control functions, such as anterior and posterior cingulate cortex; insula; and posterior, parietal, and prefrontal cortex. In order to model these regions in a comprehensive way, Menon (2011) proposed an interaction model called “Triple network model”: DMN (default mode network), CEN (central executive network), and SN (salience network) [37]. Thus, it highlights the essential role of these regions and their interactions in the control of the individual’s behavioral strategies (Figure 1). Studies of alterations in the synchronization of neural networks reflect a marked hypoactivity in these patients. The analysis of electrical neuroimaging (eLORETA) helps to understand the dysfunctions associated with the syndrome in cognitive areas [38,39,40].

The study of cerebral perfusion using arterial spin labeling (ASL), a non-invasive method that is also used with MRI, shows a decrease in regional cerebral perfusion. These studies complement volumetry and connectivity (anatomical and functional, tractography, and BOLD) [41,42,43].

The MR-spectroscopy performs a non-invasive analysis of different metabolites (N-acetylaspartate (NAA), creatine (Cr), choline (Cho), myo-inositol (MI), glutamine (Gln), glutamate (Glu), and lactate). Some authors report an intraventricular increase of lactate among them, which may be a result of the anaerobic metabolism [44,45,46]. Lactate accumulation may also appear in pathologies, such as fibromyalgia [47], non-psychotic depressive syndrome [48], and pathologies that frequently overlap with ME/CFS. Lactate is linked to mitochondrial metabolism. Several authors refer to the compromise of mitochondrial function, finding alterations in oxidative metabolism with a decrease in glutathione [35,43].

Studies with MRI (7.0-Tesla) have reported an overactive metabolism with GABA decrease in the anterior cingulate cortex and glutamate and glutamate + glutamine elevation in the putamen [35]. The neuronal viability marker N. acetylaspartate (NAA) is decreased (↓ NAA/Cr) in the prefrontal cortex, an area involved in the cognitive control. The assessment of this ratio may be useful to understand cognitive dysfunctions and to establish a subgroup of patients showing higher pain scores [49].

Neuroinflammation (encephalitis/myelitis) has been demonstrated with positron tomography (PET) [50,51] but lacks the neuropathological correlate. Other publications with this technique focus on the metabolism of serum acetylcholine (muscarinic) antibodies, trying to establish the commitment of neurotransmitters and their receptors in cognitive function, reaching the conclusion that they do not alter these functions [52].

Recently, MRI studies measuring cerebral perfusion indicated high resting cerebral perfusion associated with greater severity of dizziness symptoms. This may correspond to alterations in the brain stem neurovascular regulation mechanisms to cope with changes in blood pressure due to orthostatic stress [53,54].

The involvement of any of these brain stem regions (ascending reticular substance, among others) could explain both cognitive dysfunction and the involvement of the autonomic nervous system in the control of homeostatic mechanisms.

## 4. Dysautonomia

The autonomic dysfunction constitutes one of the most frequent features in ME/CFS [55]. Palpitations, orthostatic intolerance (hypotension, tachycardia), frequent need to urinate, alterations in thermoregulation, etc., appear in 90% of patients with ME/CFS [56]. Some authors consider the syndrome as a dysautonomic pathology [57], so much so that they come to propose it as a biomarker of the disease [58,59]. Perhaps in the future, a change in the nomenclature of the syndrome could be proposed as “idiopathic chronic dysautonomic syndrome”.

Variables related to blood pressure (systolic pressure, diastolic volume, cardiac output, heart rate, etc.) are clearly altered for many affected by ME/CFS compared to controls [60]. We can also find an inability to focus vision, hypersensitivity to light [61], noise, vibration, smell, taste, and touch, as well as alterations in depth perception, muscle weakness, spasms, poor coordination, a feeling of instability, and ataxia [18].

The COMPASS-31 is an abbreviated clinical questionnaire organized into six domains to explore the autonomic nervous system (orthostatic intolerance, vasomotor, secretomotor, pupillomotor, gastrointestinal transit, or bladder control). This questionnaire is a useful tool for the triage of patients under suspect and can guide towards the need for a deeper evaluation of the autonomic function (respiratory arrhythmia, Valsalva maneuver, tilt table protocol, or others) [62] in a dysautonomia unit, highlighting its importance.

The sympathetic/parasympathetic imbalance has been referred to in several publications that highlight the predominance of sympathetic activity. It causes alterations in heart rate variability (HRV) with an increase in the ratio (LF/HF), and provides a possible explanation of both physical and mental fatigue [63,64]. Delving into this imbalance (sympathetic/parasympathetic), according to some authors it points to a dysfunction of the autonomic nervous system with reduced nocturnal parasympathetic activity [65] and an increase of sympathetic activity [66].

In the latest proposal for a terminological change in this pathology, the US National Academy of Medicine advocates for the name of Systemic Exertion Intolerance Disease [16], highlighting an important feature of the syndrome, the marked and rapid fatigability together with a prolonged recovery time (at least 24 h). This aspect differs from the “central” fatigue experienced by patients with pathologies, such as multiple sclerosis or Parkinson’s disease. In them, rehabilitation including physical exercise is mandatory. It is a pillar in the management of these diseases, being irrelevant to the presence of post-exertional discomfort. In contrast, the “chronic status” of ME/CFS patients deteriorates or worsens significantly if physical exercise is incorporated into their treatment [67,68,69].

The biological background of ME/CFS symptomatology involves molecules distributed throughout the body. All of them are regulators of essential functions in tissue homeostasis, for instance receptors such as adrenergics [70], serotonin [71] or TRP (transient receptor potential channels) [72].

The molecular and anatomical diversity among serotonin receptors means that the serotonergic system is involved in the regulation of pain, inflammation, memory, sleep, appetite, thermoregulation, and various neuroendocrine functions, as well as depression, anxiety and chronic fatigue [73]; all or many of these symptoms are part of the syndrome that concerns us.

In this sense, Yamamoto et al. (2004) [74] reported a decrease in the density of serotonin transporter (5-HTTs) in anterior cingulate cortex and Cleare et al. (2005) [75] reported a marked decrease in 5-HT1A receptors in the hippocampus, both by positron tomography (PET). There are a considerable number of articles highlighting the essential role of neurotransmitters in ME/CFS [76], but they always offer small samples of patients and inhomogeneous clinical criteria for identification.

Several studies provide evidence on the presence in serum of autoantibodies against neurotransmitter receptors, such as acetylcholine [52,77], norepinephrine [70], and serotonin [78], at least in a subgroup of patients. In this sense, some authors postulate that we could be facing an autoimmune pathology [79], at least in some of its forms, opening the possibility for a subgroup of patients (yet to be determined) to a treatment with immunomodulators or immunoadsorption [80]. However, recent publications did not find an increase in the frequency of autoantibodies, such as NMDA (related to some autoimmune encephalitis) nor LRP4, ACHR, and MuSK (associated with myasthenia gravis) [81], nor against the mitochondrial membrane [82]. Larger longitudinal studies are needed to determine the role of much of these antibodies, since they also appear elevated in control samples.

Neuroanatomical structures involved in the symptomatology of ME/CFS are multiple. We are suggesting the hypothalamus (paraventricular nucleus) as the biological center of the symptomatology. It triggers the stimulation of the hypothalamic–pituitary–adrenal axis, receiving modulation from emotional processing regions, such as the prefrontal cortex, the hippocampus, and the amygdala, via the bed nucleus of the stria terminalis (modulator in dysregulation of mood, anxiety, and fear) [83].

Autonomic symptoms are dependent on the regulation of the paravertebral ganglia or the celiac plexus. Constipation is related to the myenteric plexuses, nocturia, impotence, or urination related to the pelvic plexus, dysesthesias, and pain depending on epidermal innervation. Deficits of attention are related to the prefrontal cortex; sleep disorders are related to the raphe nuclei and locus coeruleus, etc.; and emotional symptoms are related to the amygdala or hippocampus; much of them point to fine fiber neuropathy as a potential substrate [19].

This pathology has a cardiac, pulmonary, digestive, or bladder component, but it does not seem to be primarily cardiological, pneumological, gastroenterological, or urological, etc. However, it does reasonably allow us to think that it was originally a dysregulation of the central, peripheral, and autonomic nervous systems; it goes without saying that one of the central characteristics of the nervous system is to be highly integrated, coordinated, and reciprocally modulated. It is a syndrome in which some symptoms are explained from the neurology. Perhaps it is primarily or fundamentally a neurological disease, as it appears in the international classification.

To advance our understanding of the syndrome in the coming decades, the nomenclature and diagnostic criteria will have to be clarified, and a universal research methodology based on relevant aspects must be established, thus debuting the syndrome (allowing modeling the disease, post-viral and post-toxic, etc.), clinical phenotypes, the same evaluation scales, and standardized questionnaires, etc. The collection of this evidence will allow for the understanding of the role of the nervous system. In this sense, the EUROMENE Group (The European Network on Myalgic Encephalomyelitis/Chronic Fatigue Syndrome) has made an effort for consensus [84,85].

## 5. Conclusions

The neurobiopathological substrate of ME/CFS is unknown. There currently is no neuroimaging finding or specific laboratory test to establish the diagnosis. Changes reported in volumetry, cerebral blood flow, anatomy, and functional connectivity, at rest as well as in response to stimuli reveal the existence of brain dysfunctions, whose meaning is yet to be determined. The interpretation of findings is complicated by the lack of a consensual study protocol.

The available evidence on the involvement of the autonomic nervous system (sympathetic/parasympathetic imbalance) indicates that the neurologist plays an essential role in the clinical evaluation of the syndrome and highlights the potential benefits of dysautonomia units for a better understanding of these dysfunctional pathologies.

## Figures and Tables

**Figure 1 medicina-57-01030-f001:**
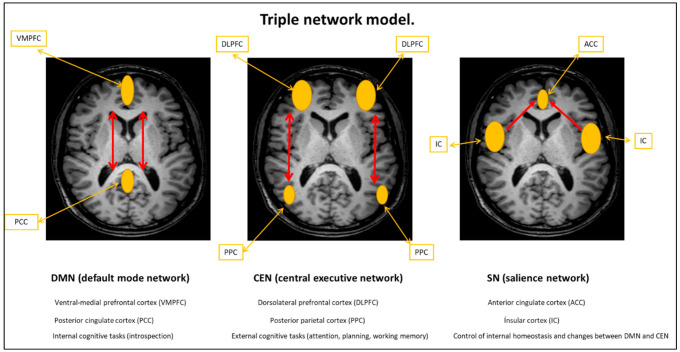
Graphic illustration of the triple network model for cognitive control proposed by Menon composed of the “Default Motor Network” (DMN), the “Salience Network” (SN), and the “Central Executive network” (CEN). According to this model, the anterior insula (belonging to the executive network) plays a key role since it activates CEN and deactivates DMN in response to outgoing stimuli to perform tasks of attention, planning, and working memory.

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
