# Peer review of "Myalgic Encephalomyelitis/Chronic Fatigue Syndrome: A Neurological Entity?"

_medicina, 2021, doi:10.3390/medicina57101030_

Round 1

Reviewer 1 Report

Dear Authors,

This is an interesting manuscript, where the authors seems to propose considering “CFS/ME” as a neurological entity – as the title suggests. According to their abstract, their methodological approach was to carry out a narrative review on publications considering studies that used neuroimaging and/or evaluation of dysautonomia. Their study was based on searching “the usual databases” for identifying the material for analysis. Their findings were presented under a topic called “Development”, where a description of post-exertion malaise (the hallmark of ME/CFS) is followed by some explanation of the “central fatigue” and its potential correlation with brain dysfunctions, involving distinct stimuli and brain areas. This was followed by a brief consideration on the main diagnostic criteria for ME/CFS – as it is referred by the main clinical diagnostic criteria, and by two topics “Neuroimaging” and “Dysautonomia”.  Under these topics, the authors managed to provide a good commentary on the studies’ findings, considering them as potential pathophysiological bases for explaining symptomology and identifying possible limitations.  Then, their narrative concludes with a statement that while neuroimaging studies do not contribute to the diagnosis – “probably due to the absence of a specific 30 protocol for the analysis”, dysautonomia studies are “highly promising” for the identification of potential biomarkers.

I consider that narrative reviews on ME/CFS can contribute to elucidate the pathophysiology of this complex disease, particularly from a those presenting specialty perspective. However, I think this manuscript could be improved by the following:

  1. To reconsider the title of the paper, or to make a comment in the conclusion by clearly addressing the question posed in the title.
  2. To provide a brief methods session, mentioning the searched databases, the search terms, and the criteria used to consider the papers, and the number of papers considered from 2006 to date.
  3. To consider the role of diagnostic criteria, for studies on ME/CFS to avoid selection bias, and to standardise the research instruments, please see:

Mudie, K.; Estévez-López, F.; Sekulic, S.; Ivanovs, A.; Sepulveda, N.; Zalewski, P.; Mengshoel, A.M.; De Korwin, J.; Hinic Capo, N.; Alegre-Martin, J.; Castro-Marrero, J.; Murovska, M.; Nacul, L.; Lacerda, E. Recommendations for Epidemiological Research in ME/CFS from the EUROMENE Epidemiology Working Group. Preprints 2020, 2020090744 (doi: 10.20944/preprints202009.0744.v1).

Nacul, L.; Authier, F.J.;Scheibenbogen, C.; Lorusso, L.;Helland, I.B.; Martin, J.A.; Sirbu, C.A.;Mengshoel, A.M.; Polo, O.; Behrends,U.; et al. European Network onMyalgic Encephalomyelitis/ChronicFatigue Syndrome (EUROMENE):Expert Consensus on the Diagnosis,Service Provision, and Care of Peoplewith ME/CFS in Europe. Medicina 2021,57, 510.  https://doi.org/10.3390/medicina57050510

Additionally, I would recommend considering the following specific references, which touch some of the points that I mentioned above, and which are missing in the current version of your manuscript:

https://www.commondataelements.ninds.nih.gov/sites/nindscde/files//Doc/Updates/MECFS_CDE_Revision_History.pdf

Komaroff AL, Takahashi R, Yamamura T, Sawamura M. Neurologic Abnormalities in Myalgic Encephalomyelitis/Chronic Fatigue Syndrome: A Review. Brain Nerve. 2018 Jan;70(1):41-54. Japanese. doi: 10.11477/mf.1416200948. PMID: 29348374.

VanElzakker MB, Brumfield SA, Lara Mejia PS. Neuroinflammation and Cytokines in Myalgic Encephalomyelitis/Chronic Fatigue Syndrome (ME/CFS): A Critical Review of Research Methods. Frontiers in Neurology. 2019 Jan; 9 : 10333. doi: 10.3389/fneur.2018.01033   

Kind Regards.   

Author Response

Reviewer 1

Comments and Suggestions for Authors.

This manuscript could be improved by the following:

  1. To reconsider the title of the paper, or to make a comment in the conclusion by clearly addressing the question posed in the title.

Authors. This aspect has been clarified in conclusions.

   2.To provide a brief methods session, mentioning the searched databases, the search terms, and the criteria used to consider the papers, and the number of papers considered from 2006 to date.

Authors. It is a narrative review, not a systematic review, where a document is prepared with the most relevant data on the discussion of the topic.

3.To consider the role of diagnostic criteria, for studies on ME/CFS to avoid selection bias, and to standardise the research instruments, please see:

Mudie, K.; Estévez-López, F.; Sekulic, S.; Ivanovs, A.; Sepulveda, N.; Zalewski, P.; Mengshoel, A.M.; De Korwin, J.; Hinic Capo, N.; Alegre-Martin, J.; Castro-Marrero, J.; Murovska, M.; Nacul, L.; Lacerda, E. Recommendations for Epidemiological Research in ME/CFS from the EUROMENE Epidemiology Working Group. Preprints 2020, 2020090744 (doi: 10.20944/preprints202009.0744.v1).

Nacul, L.; Authier, F.J.;Scheibenbogen, C.; Lorusso, L.;Helland, I.B.; Martin, J.A.; Sirbu, C.A.;Mengshoel, A.M.; Polo, O.; Behrends,U.; et al. European Network onMyalgic Encephalomyelitis/ChronicFatigue Syndrome (EUROMENE):Expert Consensus on the Diagnosis,Service Provision, and Care of Peoplewith ME/CFS in Europe. Medicina 2021,57, 510.  https://doi.org/10.3390/medicina57050510

Additionally, I would recommend considering the following specific references, which touch some of the points that I mentioned above, and which are missing in the current version of your manuscript:

https://www.commondataelements.ninds.nih.gov/sites/nindscde/files//Doc/Updates/MECFS_CDE_Revision_History.pdf

Komaroff AL, Takahashi R, Yamamura T, Sawamura M. Neurologic Abnormalities in Myalgic Encephalomyelitis/Chronic Fatigue Syndrome: A Review. Brain Nerve. 2018 Jan;70(1):41-54. Japanese. doi: 10.11477/mf.1416200948. PMID: 29348374.

VanElzakker MB, Brumfield SA, Lara Mejia PS. Neuroinflammation and Cytokines in Myalgic Encephalomyelitis/Chronic Fatigue Syndrome (ME/CFS): A Critical Review of Research Methods. Frontiers in Neurology. 2019 Jan; 9 : 10333. doi: 10.3389/fneur.2018.01033   

Authors: Added, it is included: Mudie et al., 2020, and Nacul et al., 2021.

But it is not included for the following reasons:

Komaroff et al., 2018 : It is a Japanese version, not having been found in English.

VanElzakker et al., 2019 : It focuses on methodology for the determination of cytokines.

Thank you.

Reviewer 2 Report

First, this was not a systematic review or original report and therefore, the “materials and methods” section is inappropriate. This is a hypothesis paper or opinion essay, so should not be formatted with Methods/Results/Conclusions.

Although there is a “Materials and Methods” section in the abstract, there is no methods section in the manuscript draft. Very few papers are mentioned so it should not be described as a “review”.

Also, the “Results” section states incorrectly that “all these findings” on dysautonomia are “related to the presence of auto-antibodies (igG) in serum against receptors involved in homeostatic control mechanisms.”

This conflicts with the discussion section that admits that recent publications “do not reflect an increase in the frequency of autoantibodies”, suggesting only that further research is needed.

The manuscript excludes any mention of the potential role of infections such as HHV-6, EBV, enterovirus  and COVID-19 in triggering fatigue, and the impact these infections can have on autoantibodies (see Jumah 2021 for example.) Adding a discussion on this topic would improve the manuscript.

The manuscript should be checked carefully for grammar errors. For example, the sentence beginning at line 79 is missing a verb.

On the positive side, the summary of imaging in CFS/ME is excellent the authors make important points worthy of discussion, such as their complaint about the absence of a list of comorbidities that could enable the establishment of clinical subgroups.

Author Response

Reviewer 2

Comments and Suggestions for Authors.

  1. First, this was not a systematic review or original report and therefore, the “materials and methods” section is inappropriate. This is a hypothesis paper or opinion essay, so should not be formatted with Methods/Results/Conclusions.

Although there is a “Materials and Methods” section in the abstract, there is no methods section in the manuscript draft. Very few papers are mentioned so it should not be described as a “review”.

Authors. The approach has been adjusted to the type of article; narrative review. Deleting in the abstract; "Materials, methods and results".

2. Also, the “Results” section states incorrectly that “all these findings” on dysautonomia are “related to the presence of auto-antibodies (igG) in serum against receptors involved in homeostatic control mechanisms.”

This conflicts with the discussion section that admits that recent publications “do not reflect an increase in the frequency of autoantibodies”, suggesting only that further research is needed.

Authors. The question of the syndrome as an autoimmune disease is an issue to be determined. Studies are unclear about this problem. There are authors who reflect the presence of these against; acetylcholine, norepinephrine and serotonin receptors (at line 256), however other authors do not find the presence of autoantibodies for example; (NMDA, LPR4, ACHR, MusK) (at line 261 and 262).It is also indicated at lines 263 and 264 that longitudinal studies are needed to determine their role since they also appear in healthy controls.

3.The manuscript excludes any mention of the potential role of infections such as HHV-6, EBV, enterovirus and COVID-19 in triggering fatigue, and the impact these infections can have on autoantibodies (see Jumah 2021 for example.) Adding a discussion on this topic would improve the manuscript.

Authors. This manuscript does not focus on the ethiopathogenic or pathophysiological mechanisms, therefore it does not refer to the role of infections in the syndrome. Neither the mechanisms of generation of autoantibodies. For the elaboration of the manuscript, a dense bibliography has been consulted over a year, to finally select those publications that best reflect the exposition.

4.The manuscript should be checked carefully for grammar errors. For example, the sentence beginning at line 79 is missing a verb.

Authors. The review and grammatical style of the text have been carried out by the editorial service of the magazine.

5.On the positive side, the summary of imaging in CFS/ME is excellent the authors make important points worthy of discussion, such as their complaint about the absence of a list of comorbidities that could enable the establishment of clinical subgroups.

Authors. The interpretation of the results of the different authors is complex, with very disparate methodologies and with patients who present multiple associated comorbidities.

Thanks for your comments.

Round 2

Reviewer 2 Report

The manuscript is fine as revised.